# The Role of Small Airway Disease in Pulmonary Fibrotic Diseases

**DOI:** 10.3390/jpm13111600

**Published:** 2023-11-13

**Authors:** Georgios I. Barkas, Zoe Daniil, Ourania S. Kotsiou

**Affiliations:** 1Department of Human Pathophysiology, Faculty of Nursing, University of Thessaly, 41500 Larissa, Greece; gbarkas-m@uth.gr; 2Department of Respiratory Medicine, Faculty of Medicine, University of Thessaly, 41110 Larissa, Greece; zdaniil@uth.gr

**Keywords:** autoimmune disease, fibrosis, interstitial lung disease, small airways

## Abstract

Small airway disease (SAD) is a pathological condition that affects the bronchioles and non-cartilaginous airways 2 mm or less in diameter. These airways play a crucial role in respiratory function and are often implicated in various pulmonary disorders. Pulmonary fibrotic diseases are characterized by the thickening and scarring of lung tissue, leading to progressive respiratory failure. We aimed to present the link between SAD and fibrotic lung conditions. The evidence suggests that SAD may act as a precursor or exacerbating factor in the progression of fibrotic diseases. Patients with fibrotic conditions often exhibit signs of small airway dysfunction, which can contribute to worsening respiratory symptoms and decreased lung function. Moreover, individuals with advanced SAD are at a heightened risk of developing fibrotic changes in the lung. The interplay between inflammation, environmental factors, and genetic predisposition further complicates this association. The early detection and management of SAD can potentially mitigate the progression of fibrotic diseases, highlighting the need for comprehensive clinical evaluation and research. This review emphasizes the need to understand the evolving connection between SAD and pulmonary fibrosis, urging further detailed research to clarify the causes and potential treatment between the two entities.

## 1. Introduction

Interstitial lung disease, or ILD, encompasses a variety of lung disorders that affect the pulmonary interstitium [1]. These conditions can have devastating consequences, including reduced exercise capacity, poor gas exchange, and a lower quality of life. Sadly, most patients with ILD have limited treatment options, and respiratory failure and death are often inevitable outcomes [1,2,3]. The body’s natural response to pathogens is the formation of fibrosis, but in cases of pulmonary fibrosis, excessive inflammation and fibrotic responses can occur, leading to tissue remodeling and matrix deposition. We urgently need better treatments for ILDs to improve patients’ lives and prevent these devastating outcomes.

Fibrosis is a serious issue caused by the excessive deposition of the extracellular matrix (ECM) during wound healing [4].The process of fibrogenesis is orchestrated by several types of cells and signaling mechanisms, and it can lead to a variety of health complications. It is crucial to understand that fibrosis is not a disease in itself but rather an outcome of dysregulation following many types of tissue injury [5,6]. During tissue injury, fibroblasts become activated and increase the secretion of inflammatory mediators, contractility, and the synthesis of ECM components. This process leads to a minor increase in ECM deposition, which is usually cleared from the wound site via apoptosis after injury repair [6,7]. However, in pulmonary fibrosis, myofibroblasts, resulting from fibroblasts and other mesenchymal cells, repeatedly secrete ECM components that fail to clear via apoptosis. These cells remain activated, leading to excessive ECM deposition [6,7,8]. The accumulation of ECM can disrupt normal tissue architecture, leading to lung failure, increased stiffness, cell damage, and oxygen diffusion obstructions [9]. Factors such as infectious agents, alcohol, environmental particles, gene mutations, and predispositions can cause fibrosis, while inflammation upregulates mediators of fibrosis [6,7,8,9].

Small airway obstruction, small airway dysfunction, and small airway disease (SAD) refer to pathophysiology within bronchioles and non-cartilaginous airways 2mm or less in diameter [10]. Measuring SAD function is crucial for determining pulmonary mechanics and general lung function in patients with respiratory diseases, where small airways are the predominant site of resistance [7,10]. Accessing and visualizing these smaller airways in clinical practice is complex and requires the use of non-invasive techniques such as spirometry, oscillometry, resistance measurements, plethysmography, nitrogen washout, alveolar nitric oxide, helium–oxygen flow–volume curves, and high-resolution computed tomography (HRCT) [7,8,9,10].

The use of spirometry in evaluating SAD is controversial, as the spirometric parameters used are subject to wide variation. Forced expiratory flow (FEF) indicators such as FEF50%, FEF75%, and FEF25-75%, also known as maximum mid-expiratory flow (MMEF), are used to assess SAD. If at least two of these indicators fall below the expected value of 65%, small airway dysfunction is likely [11]. Plethysmography is another method used to estimate residual air volume (RV) and RV/total lung volume (TLC), which provide information about air trapping and lung hyperinflation and are used to evaluate small airway function [12]. The nitrogen washout technique is used to calculate the lung clearance index (LCI) by recording an inert tracer gas being cleared from the lungs during normal tidal breathing. LCI is defined as the number of lung turnovers required to wash out an inert gas to 1/40th of its initial concentration [13,14]. Helium–oxygen flow–volume curves have been used since the 1970s to assess small airway dysfunction by evaluating obstruction in both the large and small airways. These tests show the increased volume of isoflow when the maximal flow is reduced due to loss of elastic recoil or an increase in upstream resistance [15,16].

Oscillometry, a noninvasive and patient-friendly technique, is the gold standard for detecting SAD. This technique involves applying pressure oscillations at the mouth to accurately measure pulmonary resistance and reactance, which are used to determine small airway dysfunction. Two methods are utilized for this purpose: the forced oscillation technique (FOT) and impulse oscillometry (IOS). The FOT transmits sound waves of different frequencies sequentially into the respiratory tract during tidal ventilation, while IOS transmits a mathematically decomposed impulse to reduce the time of the test and provides a high signal-to-noise resolution. The use of these two methods significantly reduces the likelihood of errors in measurement [13,17,18].

The small conducting airways are a crucial area within the lungs where disease can accumulate without being detected by conventional tests [19,20,21,22]. Research suggests that small airways undergo more extensive inflammation and remodeling than large airways in several lung diseases, particularly chronic obstructive pulmonary disease (COPD) and asthma [23,24,25]. In COPD, small airways’ primary role is attributed to the greater degree of airflow limitation, while inflammatory and structural changes in distal airways increase with more severe bronchial obstruction. A study [26], mainly using IOS systems, demonstrated that small airway obstruction occurs in most COPD patients, showing worse respiratory reactance and spirometry, more extensive lung hyperinflation, and poorer health states compared to samples of patients without small airway dysfunction [25,26]. In asthmatic patients, SAD is detectable across all the stages of asthma, but especially among severe forms. It is associated with poorer quality of life severity, instability, and exacerbations [27,28]. The small airways have been identified as a crucial site of inflammation in asthma. SAD contributes independently to asthma’s clinical expression [29] and has been associated with poorer asthma control in patients [30]. Half of a cohort study of adult asthma subjects showed evidence of peripheral airway involvement, and with smoking, a reduced forced expiratory volume in 1 sec (FEV1) and upregulated blood eosinophils, a “small airway asthma subtype”, has been proposed [31]. The authors suggested that smoking harms peripheral airway function [31]. The small airway dysfunction asthma phenotype is accumulating interest, with evidence supporting a distinct clinical phenotype prevalent in patients in all management guidelines and associated with poor disease control. However, other research supports the idea that small airway dysfunction might indicate early disease rather than a phenotype [32].

The role of SAD and its connection to fibrotic and interstitial pulmonary disease have not been widely reported. This review aims to shed light on the multifaceted roles of small airway dysfunction in pulmonary fibrotic diseases. For this reason, we conducted a literature search in the PubMed search engine (https://pubmed.ncbi.nlm.nih.gov, accessed on 2 August 2023) with the use of filters (publication limit for the last five years and English language) and the following keywords: fibrosis AND small airways, idiopathic lung disease AND small airway disease, small airway dysfunction AND fibrosis. We also included other publications, mostly screened through reference lists. The flowchart of the study is presented in Figure 1.

## 2. Small Airways in IPF

Idiopathic pulmonary fibrosis (IPF) is a chronic lung disease that primarily affects older adults. It is characterized by the progressive worsening of dyspnea and lung function, leading to respiratory failure, and it has a poor prognosis. IPF is fibrosing interstitial pneumonia associated with radiological and histologic features of usual interstitial pneumonia (UIP). Despite extensive research, the cause of IPF remains unknown [5,33].

The pathogenesis of IPF is complex, involving a wound-healing process in response to lung injury. The disease is characterized by the scarring of lung tissue, which reduces gas exchange and leads to respiratory failure. IPF is considered a multi-factorial disease with a complicated pathogenesis influenced by environmental, aging, and genetic factors [34]. Persistent micro-injuries cause damage to alveolar epithelial tissues in IPF, leading to aberrant repair processes. Pathologically, the pulmonary interstitium is involved, but the disease may also affect the airways, pleura, and pulmonary circulation [35].

Bronchiolitis is an injury to the bronchioles resulting in inflammation and potentially fibrosis [36]. The current pathogenic paradigm supports the idea that the disease arises from the premature senescence of alveolar epithelial cells following repetitive alveolar injury in genetically susceptible individuals [37]. This provides a start to the ECM accumulation cascade, which results in the fibrosis of the tissue. Recent studies propose that the accumulation of mucus in the distal respiratory bronchioles can lead to continuous lung tissue injury, therefore starting the accumulation cascade [38,39]. Moreover, studies have reported a significant reduction in terminal bronchioles in IPF patients, with supporting evidence indicating that the airway walls of terminal bronchioles are thickened and dilated, with distorted lumens, leading to honeycomb cysts [40,41]. While IPF mostly spares the airways, recent research has identified SAD as a possible factor influencing the disease [39,42,43].

There are only a few studies determining SAD in IPF patients’ dysfunction by using either IOS [44] or spirometric results such as MMEF, FEF50%, FEF75%, FEV1%, and HRCT results [45,46]. At the same time, older studies have found histopathological evidence of SAD in IPF patients [42,47]. Therefore, despite the heterogeneous nature of IPF, there is evidence that SAD affects IPF, and it might help manage and diagnose the disease in the near future.

## 3. The Small Airway Disease in Other Fibrotic Diseases

Interstitial pulmonary disease, caused by the thickening and scarring of lung tissue, is characterized by a restrictive defect without evidence of obstruction to airflow [48]. Interstitial pneumonias are a group of diseases defined by their distinct histopathological features, with the histological pattern being the archetype of progressive fibrosis [49,50]. Recent research has provided insights into small airway changes, including abnormal pathologic, physiological, and imaging changes, in interstitial pneumonias [51]. Most ILDs result in structural alterations of the small airways and the alveoli, which result in airway obstruction [39,40,41].

Nonspecific interstitial pneumonia (NSIP) is an ILD that can be categorized as either idiopathic or secondary due to connective tissue disease or caused by toxins or numerous other causes [49]. A study of biopsy and autopsy samples on usual interstitial pneumonia (UIP) and NSIP patients revealed that NSIP patients present increased bronchiolar and peribronchiolar inflammation, fibrosis, and decreased luminal areas, to conclude that small airways may take part in the lung remodeling process [52].

Hypersensitivity pneumonitis (HP), which is a complex syndrome characterized by inflammation and fibrosis in both the airways and the lung parenchyma, is caused by the inhalation of a variety of diverse antigens—which are mainly derived from bird protein and fungi in genetically susceptible and sensitized individuals [53,54]. According to the ATS/JRS/ALAT (2020) [55] and ACCP (2021) [56] classifications, we now distinguish “non-fibrotic HP” and “fibrotic” HP. It can progress to pulmonary fibrosis [57], while its connection to the small airways is extensive [57,58]. HP is one of the most frequent causes of distal airway disease. In several studies, patients with HP had small airway abnormalities as indicated by ultrasonic pneumography (UPG) and IOS, not found by spirometry and body plethysmography [57]. Abnormality within and/or around the small airways is a unique feature of HP that is observed in all patients with HP in histopathology [53,59]. It is associated with inflammation of the bronchioles predominantly by lymphocytic infiltrates and granuloma formation, causing bronchial obstruction [59]. In other words, in HP, there is a strong association with small airway abnormalities where patients often exhibit inflammation in the bronchioles, characterized by granuloma formation and lymphocytic infiltrates, leading to bronchial obstruction.

Another condition highly connected with the small airways is cryptogenic organizing pneumonia (COP), of unknown etiology, while organizing pneumonia (OP) has a known etiology [60,61]. It obstructs or erases the small airways, leading to small airway obstruction, as indicated by spirometry and histopathology [62]. The disease is influenced by several factors, including preexisting factors and genetic variants [63]. It is thought to be caused by an initial insult in the lower airways, which leads to inflammation and tissue injury. The pathogenesis of COP is not yet fully understood, but the histopathological features show injured and inflamed small airways, leading to excessive fibroproliferation due to aberrant tissue repair. Many factors have been implicated in the development of COP, including an increased level of circulating CD4+ T-cells, CD 8+ T-cells, and Th-cells. It is also hypothesized that pulmonary antigens cause cytotoxic T-lymphocytes to target endothelial cells, leading to tissue fibrosis [62,63,64,65,66,67,68,69].

To summarize, fibrotic lung diseases cause the thickening and scarring of lung tissue, which can impact airflow and lung function. Although there is a shared attribute with ILDs where lung volumes are reduced without airflow obstruction, small airways play a role in lung remodeling in the NSIP pattern. Most ILDs result in structural changes in small airways [70] and alveoli. Additionally, HP can cause inflammation to the bronchioles, leading to bronchial obstruction. It is important to note that COP or OP are highly linked to small airways.

## 4. The Small Airways in Autoimmune Interstitial Lung Disease

Apart from the known ILDs, small airways are also linked to other respiratory diseases falling under different categories, including occupational disorders and connective tissue diseases. Connective tissue disease (CTD)-associated interstitial lung disease (CTD-ILD) is the most common pulmonary manifestation of CTD, affecting the airways, lung parenchyma, and pleura while also being related to increased morbidity and mortality [71,72,73,74]. However, the connection of small airways to CTD-ILD has not been researched enough. A study of pulmonary function tests (PFTs) in CTD-ILD patients determined MMEF, FEF 50%, and FEF 75% by spirometry and made remarks regarding SAD in CTD-ILD. The study concluded that around half of the CTD-ILD patients had SAD. Moreover, there was an improvement in forced vital capacity (FVC), but there was no post-treatment improvement in SAD in CTD-ILD patients [73].

Autoimmune diseases are a group of at least 80 illnesses that share a common pathogenesis, which is an immune-mediated attack on the body’s organs [74]. The immune system is developed to protect hosts from infectious agents; however, it can lead to disease either by an inability of one or more of its components to respond protectively to a pathogen or by the failure to distinguish self from non-self, which is the basis for autoimmune diseases [75]. The etiology of autoimmune diseases is multifactorial, with genetic, environmental, hormonal, and immunological factors considered essential in their development. However, the onset of at least 50% of autoimmune disorders has been attributed to “unknown trigger factors” [76]. Some autoimmune diseases that can have pulmonary manifestations include rheumatoid arthritis (RA), systemic lupus erythematosus (SLE), and Sjögren’s syndrome (SS). RA and SLE are among the most researched due to their prevalence and the significant impact they can have on multiple organ systems. Both have been shown to have connections to small airways.

Rheumatoid arthritis (RA) is an autoimmune disease that has been recognized as a clinical entity for over two centuries [76]. It is the most frequent inflammatory arthropathy [77], characterized by chronic, systemic, and inflammatory manifestations that affect connective tissue [78,79,80,81]. The autoimmune disorder affects approximately 1% of the population worldwide [81,82]. Pulmonary involvement is one of the common causes of morbidity in patients with RA.

ILD is the most common and severe manifestation of RA lung diseases associated with parenchymal rheumatoid nodules [83]. Bronchiectasis is also a common lung characteristic of RA with an estimated prevalence of 10 and 30%, depending on the population analyzed and the imaging methodology used for detection. However, RA directly affects all the compartments of the thorax, including the lung parenchyma, airways, pleura, and less common vessels [84].

The FEF25, FEF50, FEF75, and FEF25-75 measurements in 99 patients with RA, in the case–control study of Zohal MA et al., revealed that the abnormal results of PFTs in rheumatoid disease were higher than usual [85]. Respiratory system involvement occurs in 30–40% of RA patients, making it the second leading cause of death in patients with RA [86]. Older studies have confirmed the existence of a subgroup of non-smoking patients with RA who have isolated small airway obstruction, as indicated by FEV1, FVC, FEV1/FVC, FEF25–75, and diffusing capacity for carbon dioxide (DLCO) measurements [87], and imaging evidence of airway involvement [88]. Moreover, in patients with RA, other disorders with close connections to the airways—like OP—have been supported [89]. Although newer studies support that small airway dysfunction is present in all patients with RA, there is a study by Singh R et al. documenting that small airway dysfunction was present only in one-third of the patients with RA, especially in those with a short duration of disease, low to moderate disease activity, and no respiratory symptoms as indicated using spirometry, the forced oscillation technique (FOT), and HRCT of the chest [90]. Another supported the finding that small airway disorder in RA patients has been significantly associated with abnormal FEF25-75, respiratory symptoms, longer smoking history, and disease duration [91]. Hence, RA is highly connected to the small airways’ obstruction, dysfunction, and complete obliteration; however, other factors like smoking history can be harmful and aggravate lung disease.

SLE and SS are two chronic autoimmune inflammatory disorders that impact the lungs and can lead to serious morbidity and mortality [92]. While there is no cure for SLE, it can be managed effectively with medication. However, the mortality rate is still high, with renal disease, cardiovascular disease, and infection being the most common causes of death among patients [93]. The exact cause or underlying pathophysiological mechanisms that trigger the autoimmune response in these diseases remain largely unknown, although researchers have suggested the role of gene susceptibility. It is believed that a combination of susceptibility genes, the absence of protective genes, and epigenetics may all contribute to the development of these conditions [92,93,94]. Data from older studies indicate a very high prevalence of pulmonary function abnormalities in SLE patients [95], obstructive airway disease, and the rapid deterioration of the respiratory function determined by low PFTs, i.e., FVC, DLCO, and FEV1/FVC ratio [96], or by using, in addition to PFTs, chest X-rays, thoracic CT scans, ventilation–perfusion (VQ) scanning, and plethysmography with TLC measurement, while also identifying other pulmonary features such as pleural disease, pulmonary nodules, pulmonary cysts, and OP [96,97]. Studies conducted on juvenile SLE (jSLE) patients have found reduced values for forced vital capacity (FVC) and forced expiratory flow (FEF) 25–75%. However, the ratio of forced expiratory volume in one second (FEV1) to FVC was similar to that of healthy individuals. As a result, jSLE patients were found to have a significant restrictive pattern and small airway involvement [98,99]. Bronchiolitis has been linked to lupus in various case reports [100,101,102,103,104]. In female SLE patients, decreased DLCO in the lungs has also been observed, along with alveolar involvement [104,105]. In some age groups, OP has been identified as the sole pulmonary manifestation of SLE [106].

SS is a connective tissue disease that affects the exocrine glands, leading to their dysfunction and eventual destruction. This disease frequently affects the small airways, as indicated by spirometry and chest HRCT imaging, and is associated with mild to severe respiratory symptoms [105,107]. Lung involvement in SS is well characterized and is observed during the disease, although pleura involvement is more commonly seen in SLE patients. On the other hand, primary SS (pSS) typically involves the airways [108,109,110]. ILD involvement and the coexistence of small airway lesions in pSS-interstitial lung disease has been revealed through several studies of pSS patients using spirometry and total lung capacity, as well as diffuse lung capacity that were impaired [111,112]. In a cross-sectional study aimed at estimating the prevalence of chronic respiratory symptoms in pSS, 114 consecutive patients were investigated with PFTs and chest HRCT on inspiratory and expiratory phases. The study found that the most commonly recognized pulmonary disorder among symptomatic pSS patients (one-fifth of the study group) was SAD, followed by xerotrachea and interstitial lung disease [113]. Another retrospective study reported that among 14 patients who underwent surgical lung biopsy, constrictive changes were the most significant determinant of physiological and imaging presentation associated with Sjögren’s syndrome [113]. Patients with SS typically exhibit evidence of both airway and interstitial lung diseases, as confirmed by radiographs and pathology [97,114]. Even in older studies, SAD has been reported as a common manifestation of SS, with evidence of small airway obstruction in patients with pSS, supporting diffusion impairment and airway obstruction using DLCO and FEV1 measurements, respectively [115,116,117]. In a study of 61 non-smokers with pSS who underwent clinical, physiological, and radiological evaluation, and 13 of whom underwent a bronchial and/or transbronchial biopsy, findings revealed that airway epithelia were the main target of the inflammatory lesions, leading to physiological abnormalities of the obstructive small airway [95]. Meanwhile, another study in pSS patients that analyzed histopathologic findings, radiologic findings, and lung function tests reported that histopathologic patterns of pSS-ILD included lung interstitial involvement, small airway involvement, or both [118]. An eight-year follow-up study on airway hyperresponsiveness in patients with pSS, where lung function, standard spirometry, and measurements of lung volumes, diffusing capacity, and AHR to methacholine were performed, concluded that one-third of the patients developed a significant reduction in lung function. Small airway obstruction and airway hyperresponsiveness were associated with a reduction in vital capacity (VC) and the development of impaired diffusing capacity as a sign of interstitial lung disease [118,119].

Sarcoidosis is a systemic inflammatory granulomatous disease of unknown etiology that affects 2 to 160 people per 100,000 worldwide and can involve any organ [120]. It is considered an autoimmune disease, with growing research suggesting similar patterns of cellular immune dysregulation seen in other autoimmune diseases like RA. Recent large-scale population studies show that sarcoidosis frequently co-presents with other autoimmune diseases [121]. However, the pathophysiology of the disease remains poorly understood, and known autoantibodies or useful serologic markers for the diagnosis and monitoring of autoimmune disease activity are lacking [122]. Sarcoidosis is closely linked to environmental factors, especially occupational exposures such as silica, pesticides, and mold or mildew, which are associated with increased odds of pulmonary sarcoidosis [123]. Its etiology remains undetermined, characterized by variable clinical presentations and disease course [124]. Sarcoidosis can affect the airways at any level, and when the involvement includes small airways, it can resemble more common obstructive airway diseases, such as asthma and chronic bronchitis [124].

Sarcoidosis is one of the few ILDs that can affect the entire length of the respiratory tract, from the nose to the terminal bronchioles, and it causes a broad spectrum of airway dysfunction [124]. Peripheral airway obstruction may be caused by the formation of granulomas in a perilymphatic distribution along the bronchovascular bundles [124,125]. Older studies suggest that functional abnormalities in small airways may occur early in sarcoidosis [126]. Patients with pulmonary sarcoidosis who have obstructive airway disease may also have regional air trapping, which indicates SAD and can be visualized on expiratory HRCT and newer imaging modalities [124,125]. Moreover, a histopathologic study on end-stage fibrotic sarcoidosis patients investigating airways’ morphologic features found an increased number of airways, an increased airway diameter of distal airways, as well as fewer terminal bronchioles per milliliter and the total number of terminal bronchioles in all forms of fibrotic sarcoidosis, compared to controls [125]. Small airway loss is an important aspect of the pathophysiology of pulmonary sarcoidosis [126]. Other research regarding alveolar macrophages in sarcoidosis [127,128] showed that small airway macrophages are immunologically more active, which may be important for airway inflammation.

Therefore, sarcoidosis may be linked to SAD, but further investigation with new technologies for assessing SAD, such as IOS, is needed to support these findings. Granulomatosis affects the airways and leads to obstruction, inflammation, and airway remodeling, as visualized in fibrotic sarcoidosis patients. Moreover, sarcoidosis patients present with the substantial loss of distal airways, indicating a close connection of the disease to bronchioles.

In summary, autoimmune diseases share a common underlying feature: the immune system’s misguided assault on the body’s organs. RA and SLE demonstrate a strong correlation to SAD. pSS, primarily targeting exocrine glands, intricately involves the small airways, exacerbating morbidity. Sarcoidosis affects the number, diameter, and level of obstruction of the small airways. These links underline the complex relationship between autoimmune diseases and the delicate small airways, warranting further investigation for enhanced therapeutic strategies.

## 5. Future Directions, Limitations, and Research Gaps Regarding the Small Airways

Evidence shows that the small airways play a significant role in pulmonary fibrotic diseases. However, further research is needed to fully understand the role of SAD in pulmonary fibrotic disorders. Regarding future directions, the most important one is the improvement of imaging techniques to visualize and assess small airways in fibrotic diseases so that more detailed information about their structural changes can be obtained. The current imaging techniques might not capture the structural changes and remodeling happening in the quiet part of the airways, leading to an incomplete picture. IOS and forced oscillometry are promising tools to reveal that SAD should be incorporated further into daily clinical practice.

Furthermore, identifying specific biomarkers related to the involvement of small airways could be helpful for early diagnosis, disease monitoring, and the development of targeted therapies. Targeted therapeutic strategies for airway remodeling in fibrotic diseases could improve patient outcomes and quality of life. Hence, a more personalized approach tailored to the individual characteristics, including SAD, can lead to more effective interventions.

Moreover, there is also a need for more comprehensive molecular and genetic studies to understand the underlying pathogenesis of fibrotic diseases in the context of small airways. This might involve a more detailed investigation of genetic susceptibility, gene–environment interactions, and epigenetic influences. Lastly, conducting long-term studies to track the progression of small airway involvement in fibrotic diseases can provide valuable insights into disease mechanisms and treatment strategies. Research is also needed to elucidate how existing and emerging treatments for fibrotic diseases affect the small airways and whether targeted interventions are beneficial. Furthermore, more research is needed to understand the impact of environmental and lifestyle factors, such as smoking, air pollution, and occupational exposures, on developing and progressing fibrotic diseases in small airways.

## 6. Conclusions

In conclusion, SAD significantly contributes to lung function decline in various fibrotic respiratory conditions (Table 1). Diagnosing SAD can be challenging, as no specific clinical findings exist. However, a combination of methods can be used to assess the quiet zone of the lungs. SAD worsens the prognosis, and several treatments focusing on SAD (such as bronchodilators, inhaled corticosteroids, and immunomodulatory therapies) can help to improve symptoms and slow the disease’s progression. It is important to note that the small airways are a complex and dynamic system with still not fully understood mechanisms. The heterogeneous nature of SAD in fibrotic diseases certainly does not make treating and diagnosing it any easier. Therefore, treatment and diagnosis must be tailored to the individual patient with a personalized and holistic approach, impacting disease progression. Environmental factors such as smoking can contribute to and exacerbate SAD. Patients with SAD can live a long and active life with the necessary changes in their lifestyle, like quitting smoking and exercising regularly. There is a need for more research regarding the impact of SAD in fibrotic respiratory diseases with advancements in imaging instruments and treatment plans, as well as longitudinal studies to observe disease progression and its impact on patients’ lives.

## Figures and Tables

**Figure 1 jpm-13-01600-f001:**
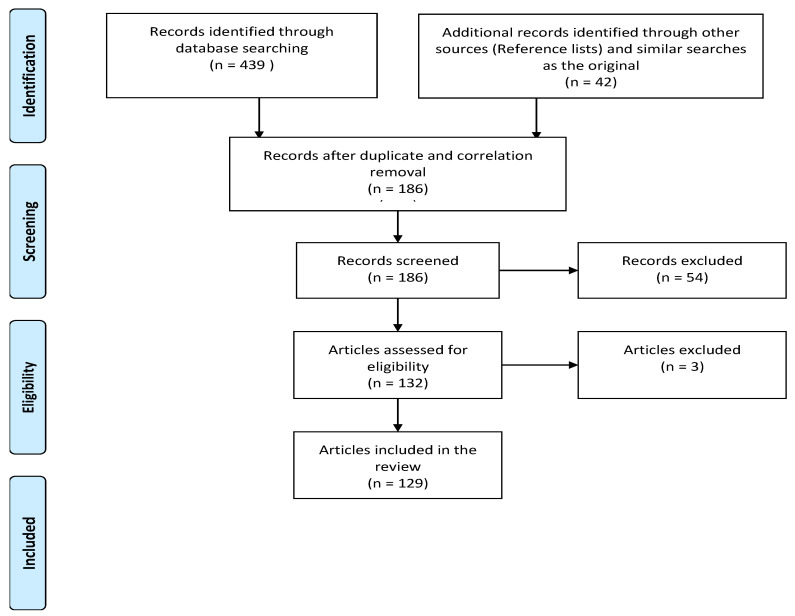
The flowchart of the study.

**Table 1 jpm-13-01600-t001:** Studies supporting the correlation of SAD and fibrotic lung diseases.

Author/Ref.	Study Design	Study Population	Main Findings
Chen H. et al. [16]	Retrospective study	No study population; data from PFTs were used	If small airway disease is present, the flow during the late expiratory phase will significantly decrease, creating a more acute angle or a smaller ∠ABC.
Verleden S. et al. [39]	Retrospective cohort study	Explanted lungs from patients with severe IPF (*n* = 11) who underwent lung transplantation were compared to a cohort of unused donor lungs—controls (*n* = 10). The control donor lungs had no known lung disease, comorbidities, or structural lung injury, and were considered suitable for transplantation.	Small airway disease is a feature of IPF, with significant loss of terminal bronchioles occurring within regions of minimal fibrosis.
Ikezoe K. et al. [41]	Experimental study	Explanted lungs from patients with IPF (*n* = 8) and donor control subjects (*n* = 8) were inflated with air and frozen.	In IPF without microscopic fibrosis, terminal bronchioles decreased in number and increased in wall area. In IPF with microscopic fibrosis, terminal bronchioles had thicker airway walls and dilated airway lumens, leading to honeycomb cysts.
Song D. et al. [51]	Experimental study	The study involved 46 patients with IIPs (19 IPF, 14 NSIP, and 13 COP). Prior to lung biopsy, pulmonary function and HRCT were assessed. Lung biopsy tissue was stained with hematoxylin–eosin to examine small airway abnormalities in pathology. The small airway abnormalities in pathology, pulmonary function, and HRCT were then compared.	The patients with IPF, NSIP, and COP had abnormal pathologic, physiological, and imaging changes in small airways.
Figuerira de Mello G.C. et al. [52]	Experimental study	Lung biopsies from 29 patients with UIP and 8 with NSIP. Biopsies were compared with lung tissue from 13 patients with CB as positive controls and 10 normal autopsied control lungs.	The UIP, NSIP, and CB patients all were found to have increased bronchiolar inflammation, peribronchiolar inflammation, and fibrosis and decreased luminal areas. The UIP patients had thicker walls due to an increase in most airway compartments. The NSIP patients presented increased epithelial areas, whereas patients with CB had larger inner wall areas. Therefore, the small airways are indeed pathologically altered and can remodel the lungs in idiopathic interstitial pneumonias.
Dalphin, J.C.; Didier, A. [59]	Literature review	No study population	HP is linked with bronchiole inflammation, granuloma formation, SAD pattern, and bronchial obstruction.
Vacchi, C. et al. [71]	Literature review	No study population	ILD is one of the most severe clinical manifestations of CTDs with the heterogeneity of clinical expression of lung involvement in CTDs, and the unpredictable disease course could contribute to the difficulties in planning RCTs.
Xu, L. et al. [73]	Two-part retrospective and prospective study	Retrospective study included patients with a diagnosis of CTD-ILD, *n* = 491, and a prospective study included *n* = 139 CTD-ILD patients.	About 50% of CTD-ILD patients had SAD. Improvement in FVC and no post-treatment improvement in SAD in CTD-ILD patients was found.
Zohal, M.A. et al. [85]	Case–control study	Ninety-nine patients with RA. History taking, physical examination, laboratory tests, and spirometry were performed. Disease Activity Score 28 (DAS28) was used to assess RA severity.	Small airway involvement was found in patients with RA.
Wright, G.D. et al. [87]	Retrospective study	Fifty-four RA patients had PFTs and chest radiography. The pulmonary function tests included FEV_1_, FVC, FEV_1_/FVC, FEF_25–75_, and TLCO.	This study confirmed the existence of a subgroup of non-smoking patients with RA who have isolated small airway obstruction. The abnormalities seemed to be progressive.
Perez, T. et al. [88]	Prospective study	Fifty patients with RA were evaluated (nine males and forty-one females; mean age: 57.8 yr), including thirty-nine nonsmokers and eleven smokers (mean cigarette consumption: 15.3 pack-yr) without radiographic evidence of RA-related lung changes.	The data collected from this study support the concept of a high incidence of airway involvement in RA patients. Evidence of airway involvement on PFTs and/or HRCT scans was correlated with respiratory symptoms.
Singh, R. et al. [90]	Retrospective study	Fifty consecutive patients were included and subjected to PFT and FOT. Those with features of SAD on PFT were subjected to HRCT of the chest.	SAD was present in one-third of the patients with RA, even in those with a short duration of disease, low to moderate disease activity, and no respiratory symptoms.
Papiris, S.A. et al. [95]	Retrospective study	Sixty-one non-smoking patients were evaluated. A bronchial and/or transbronchial biopsy was performed on 13 of the patients. Physiological data were compared with that of a control group of 53 healthy non-smoking matched for age and sex subjects.	The airway epithelia seem to be the main target of the inflammatory lesion of the lung in patients with pSS.
Karim, M.Y. et al. [97]	Retrospective/prospective study	Five patients were chosen retrospectively and two were chosen prospectively. All SLE patients.	SAD represents a rare complication of SLE.
Lin, W. et al. [112]	Retrospective study	Data of 333 patients with newly diagnosed pSS were analyzed.	ILD involvement in pSS is a common clinical occurrence
Shi, J. et al. [118]	Retrospective study	Fourteen pSS patients who underwent surgical lung biopsy were reviewed and histopathologic findings, radiologic findings, and lung function tests were analyzed.	The histopathologic patterns of pSS-ILD included lung interstitial involvement and small airway involvement or both.
Ludviksdottir, D. et al. [119]	Prospective study	An eight-year study of lung function in 15 patients who fulfilled the Copenhagen criteria for pSS. Spirometry, plethysmography, diffusing capacity, and AHR to methacholine were performed.	One-third of the pSS patients developed a significant reduction in lung function. Small airway obstruction and AHR were associated with VC reduction and impaired DLCO.
Verleden S.E. et al. [125]	Retrospective study	Airway morphology was investigated in 7 explant lungs with end-stage fibrotic sarcoidosis via CT scanning (large airways), micro-CT scanning (small airways), and histologic examination and compared with 7 unused donor lungs.	The large airways were involved differentially in subtypes of sarcoidosis, but the terminal bronchioles were universally lost.

## Data Availability

Not applicable.

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
