# Peer review of "The Role of Small Airway Disease in Pulmonary Fibrotic Diseases"

_jpm, 2023, doi:10.3390/jpm13111600_

Round 1
Reviewer 1 Report
Comments and Suggestions for Authors
This review article highlights the need to understand the evolving connection between small airway disease and pulmonary fibrosis, urging further detailed research to clarify the causes and potential treatment between the two entities. This study may provide some useful information on the role of small airway disease in pulmonary fibrotic diseases. I have some comments.
<Comments>
1. In line 91, Please describe the full term of the abbreviation, “COPD”.
2. In line 106, Please describe the full term of the abbreviation, “FEV1”.
3. In line 208, Please describe the full term of the abbreviation, “FVC”.
4. In line 248 -258, “Connective Tissue Disease-Associated Interstitial Lung Disease (CTD-ILD) is the most common pulmonary manifestation of CTD, involving multiple manifestations of respiratory complications affecting the airways, lung parenchyma, and pleura while also associated with significantly increased morbidity and mortality [59–61]. Although most patients with CTD-ILD experience stable or slowly advancing ILD, a small yet significant group exhibits a more severe and progressive course [61]. The connection ~ after treatment [62].” is completely overlapping with the previous sentence (line 199 – 209). Please modify it.
5. In line 271, “Sjögren's Syndrome”
Please correct to Sjögren's Syndrome (SS).
6. In line 304, “Sjorgen syndrome (SS)”
Please correct to “SS”
Comments on the Quality of English LanguageMinor editing of English language required.
Author Response
Response to Reviewer 1 Comments
- Point 1. This review article highlights the need to understand the evolving connection between small airway disease and pulmonary fibrosis, urging further detailed research to clarify the causes and potential treatment between the two entities. This study may provide some useful information on the role of small airway disease in pulmonary fibrotic diseases. I have some comments.
Response 1. We sincerely thank you for your kind words about our paper. We are delighted to receive positive feedback from you.
- Point 2. In line 91, Please describe the full term of the abbreviation, “COPD”.
Response 2. We have added the full term of the abbreviation.
- Point 3. In line 106, Please describe the full term of the abbreviation, “FEV1”.
Response 3. Thank you for this comment. We have added the full term.
- Point 4. In line 208, Please describe the full term of the abbreviation, “FVC”.
Response 4. The full term has now been added.
- Point 5. In line 248 -258, “Connective Tissue Disease-Associated Interstitial Lung Disease (CTD-ILD) is the most common pulmonary manifestation of CTD, involving multiple manifestations of respiratory complications affecting the airways, lung parenchyma, and pleura while also associated with significantly increased morbidity and mortality [59–61]. Although most patients with CTD-ILD experience stable or slowly advancing ILD, a small yet significant group exhibits a more severe and progressive course [61]. The connection ~ after treatment [62].” is completely overlapping with the previous sentence (line 199 – 209). Please modify it.
Response 5. Thank you for the comment. We deleted the duplicate element.
- Point 6. In line 271, “Sjögren's Syndrome”. Please correct to Sjögren's Syndrome (SS).
Response 6. Thank you, the abbreviation has been added.
- Point 7. In line 304, “Sjorgen syndrome (SS)”
Please correct to “SS”
Response 7. Thank you. We corrected it.
We really thank you for taking the time and energy to help us improve this paper.
Reviewer 2 Report
Comments and Suggestions for Authors
The article by Barkas and colleagues: “ The role of small airway disease in pulmonary fibrotic disease” concerns a very interesting and not well investigated topic. There are however some major issues that need to be addressed before it can be considered for publication.
1. The study design is unclear. The Authors do not mention how many articles they did find and how many were included in the final analysis. Flow chart would be helpful.
2. The Authors give no definition of the small airways disease or the tests used specifically in the publications they cite to characterize the small airways involvement. Again the graphical depiction (table, graph) would make the paper more comprehensible
3. Text editing is mandatory – parts of the text are in duplicate
4. English language editing is strongly recommended
Comments on the Quality of English LanguageModerate English language editing is required. Some sentences are incomprehensible. Eg page 3, verse 147-148
page 9 verses 429-434
Author Response
Response to Reviewer 2 Comments
Point 1. The article by Barkas and colleagues: “The role of small airway disease in pulmonary fibrotic disease” concerns a very interesting and not well investigated topic. There are however some major issues that need to be addressed before it can be considered for publication.
Response 1. We sincerely thank you for the kind words, the constructive feedback and your suggestions. We are delighted to receive positive feedback from you.
Point 2. The study design is unclear. The Authors do not mention how many articles they did find and how many were included in the final analysis. Flow chart would be helpful.
Response 2. Thank you for this direction, the flow chart of the study has been added.
Point 3. The Authors give no definition of the small airways disease or the tests used specifically in the publications they cite to characterize the small airways involvement. Again the graphical depiction (table, graph) would make the paper more comprehensible
Response 3. Thank you for this direction. The definition of SAD has been added in page 2, lines 53-58. In the revision, we refer the tests used to characterize the small airways involvement in the publications we cite. A graph has been added as well as a table with information on the main studies we used on the manuscript which support our theory.
Point 4. Text editing is mandatory – parts of the text are in duplicate.
Response 4. We really thank you for this point. In the revision, we have changed the parts of the texts which were duplicate (lines 207-209; 260-262 ). We really apologize for the confusion.
Point 5. English language editing is strongly recommended
Response 5. Thank you for this direction. English editing has been conducted on the paper; we hope the revised version is up to your standards.
Point 6. Moderate English language editing is required. Some sentences are incomprehensible. Eg page 3, verse 147-148; page 9 verses 429-434;
Response 6. Thank you for these points. English editing was made, as suggested.
We appreciate all your insightful comments. We found them quite useful as we approached our revision. We are grateful for the time and energy you expended on our behalf.
Round 2
Reviewer 1 Report
Comments and Suggestions for Authors
Thank you for your response.
The revised version was improved with well-prepared response letter to most comments raised by reviewers.
Although the revision for papers is not enough for my comments, the authors put much effort to revise manuscripts.
Thank you for your effort.
Minor editing of English language required.
Author Response
RESPONSE TO REVIEWER 2 COMMENTS
1. The revised version was improved with well-prepared response letter to most comments raised by reviewers. Although the revision for papers is not enough for my comments, the authors put much effort to revise manuscripts.
Thank you for your effort. Minor editing of English language required.
RESPONSE: I acknowledge that the revised manuscript has undergone a thorough editing process to ensure logical coherence and eliminate any typos. I hope that the revisions meet your standards.
Reviewer 2 Report
Comments and Suggestions for Authors
The Authors greatly improved their work. There are however some minor concerns that need to be addressed
verse 193: There is no more "chronic HP". According to the ATS/JRS/ALAT (2020) and ACCP (2021) classifications we distinguish now "non-fibrotic HP" and "fibrotic" HP
IOS/FOT the Authors need to mention FOT as one of the techniques used for the assessment of the small airways. In verse 153 they mention IOS (which in fact is a variation of FOT) and then suddenly in the verse 285 FOT appears in one of the studies. It might be confusing for the readers not familiar with pulmonary function tests
verse 346 - FEV1 does not provide any information on the diffusion impairment
Again thorough editing is required (both English language and general meaning,) keeping the Readers' comfort in mind. For example, what does the sentence: Regarding the small airways, substantial indirect evidence suggests small airwaysinvolvement." (verse 378) mean?
Comments on the Quality of English LanguageEnglish language editing is required
Author Response
RESPONSE TO REVIEWER’S 2 COMMENTS
- The Authors greatly improved their work. There are however some minor concerns that need to be addressed. verse 193: There is no more "chronic HP". According to the ATS/JRS/ALAT (2020) and ACCP (2021) classifications we distinguish now "non-fibrotic HP" and "fibrotic" HP
Response: Thank you for the valuable comment. We revised the manuscript according to your recommendation.
- IOS/FOT the Authors need to mention FOT as one of the techniques used for the assessment of the small airways. In verse 153 they mention IOS (which in fact is a variation of FOT) and then suddenly in the verse 285 FOT appears in one of the studies. It might be confusing for the readers not familiar with pulmonary function tests
Response: Thank you for this recommendation. We revised the draft accordingly (page 2, lines 74 -84).
- verse 346 - FEV1 does not provide any information on the diffusion impairment
Response: Thank you for the point. The phrase has been corrected (page 7, line 318)
- Again thorough editing is required (both English language and general meaning,) keeping the Readers' comfort in mind. For example, what does the sentence: Regarding the small airways, substantial indirect evidence suggests small airwaysinvolvement." (verse 378) mean?
Response: We appreciate your comment. I want to inform you that the revised manuscript has undergone a thorough editing process to ensure logical coherence and eliminate any typos.